# CoLT5: Faster Long-Range Transformers with Conditional Computation

**Joshua Ainslie**,[*] **Tao Lei, Michiel de Jong, Santiago Ontañón**
**Siddhartha Brahma, Yury Zemlyanskiy, David Uthus, Mandy Guo**
**James Lee-Thorp, Yi Tay, Yun-Hsuan Sung, Sumit Sanghai**

Google Research

## Abstract

Many natural language processing tasks benefit from long inputs, but processing long documents with Transformers is expensive -- not only due to quadratic attention complexity but also from applying feedforward and projection layers to every token. However, not all tokens are equally important, especially for longer documents. We propose CoLT5, a long-input Transformer model that builds on this intuition by employing conditional computation, devoting more resources to important tokens in both feedforward and attention layers. We show that CoLT5 achieves stronger performance than LONGT5 with much faster training and inference, achieving SOTA on the long-input SCROLLS benchmark. Moreover, CoLT5 can effectively and tractably make use of extremely long inputs, showing strong gains up to 64k input length.

## 1 Introduction

Many natural language processing tasks, such as summarization (Cohan et al., 2018) or question answering over long documents (Joshi et al., 2017), require machine learning models to encode long-form text. Processing long documents with a Transformer model is computationally expensive, both because attention cost scales quadratically with input length and because feedforward and attention projection layers have to be applied to each input token.

Over the past few years, many "efficient Transformer" approaches have been proposed that reduce the cost of the attention mechanism over long inputs (Child et al., 2019; Ainslie et al., 2020; Beltagy et al., 2020; Zaheer et al., 2020; Wang et al., 2020; Tay et al., 2021; Guo et al., 2022). However, especially for larger models, the feedforward and projection layers actually make up the majority of

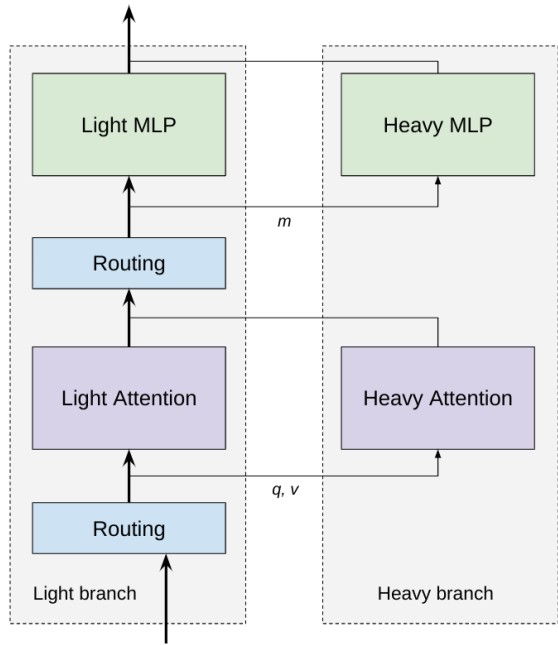

Figure 1: An overview of a CoLT5 Transformer layer with conditional computation. All tokens are processed by light attention and MLP layers, while $q$ routed query tokens perform heavier attention over $v$ routed key-value tokens and $m$ routed tokens are processed by a heavier MLP.

the computational burden and can render processing long inputs intractable.

This paper presents CoLT5 (Conditional LongT5), a new family of models that, building on top of LONGT5 (Guo et al., 2022), enables fast processing of long inputs by combining architecture improvements for both attention and feedforward layers. CoLT5 is based on the intuition that some tokens are more important than others, and we can achieve better quality for lower cost by devoting more computation to important tokens. Moreover, the fraction of important tokens is likely to diminish with document length, allowing for tractable processing of long documents.

In particular, CoLT5 divides each feedforward layer and each attention layer into a *light branch*

---

[*]Author contributions are outlined in Appendix A. Correspondence author: jainslie@google.com.

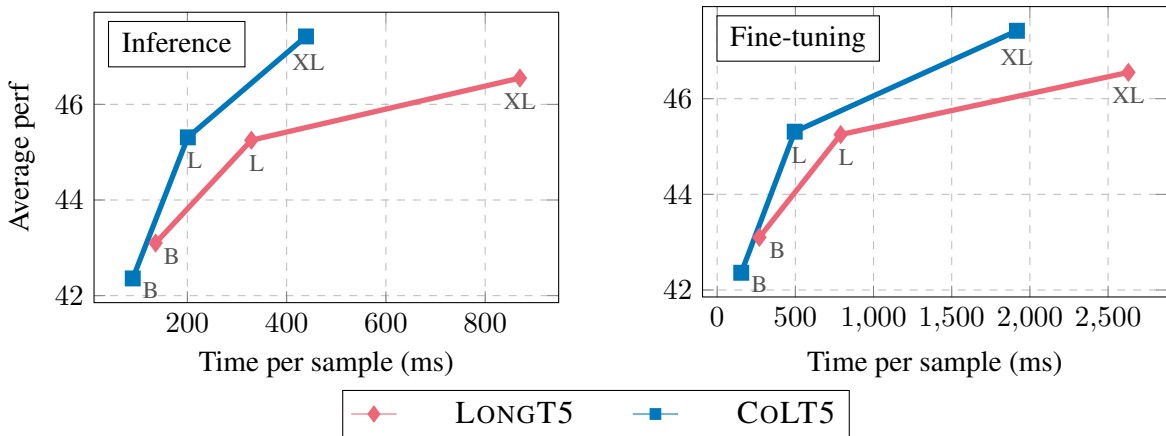

Figure 2: **COLT5 achieves stronger performance than LONGT5 at any speed.** Average performance on all datasets as a function of inference and fine-tuning time per sample (ms) for LONGT5 and COLT5 Base, Large, and XL models. LONGT5 does not use MQA, but we report speed as though it had for a conservative baseline.

which is applied to all tokens and a *heavy branch* which is applied to a set of important tokens, selected specifically for that input and component. The light feedforward branch has lower hidden dimension than standard LONGT5 while the heavy feedforward branch has higher hidden dimension. The light attention branch has fewer heads and applies only local attention, while the heavy attention branch performs full attention over another separately selected set of important tokens. Figure 1 provides an overview of the COLT5 conditional mechanism.

Finally, COLT5 also includes two other modifications to the LONGT5 architecture. COLT5 adds multi-query cross-attention (Shazeer, 2019), significantly speeding up inference. COLT5 also employs the UL2 (Tay et al., 2022) pre-training objective, which we demonstrate allows for in-context learning over long inputs.

We show that COLT5 performs much faster fine-tuning and inference with similar or better model quality, improving over LONGT5 on arXiv summarization (Cohan et al., 2018) and TriviaQA question answering (Joshi et al., 2017) datasets and achieving SOTA on the SCROLLS benchmark (Shaham et al., 2022). Moreover, COLT5 achieves further gains in quality and speed for tasks with extremely long inputs (64k tokens), with less-than-linear scaling of "focus" tokens.

## 2 Background

**Transformer FLOPs**   COLT5 follows an extensive line of work in attempting to reduce the computational cost of Transformer models, particularly over long inputs. The computational burden of Transformer models has several distinct elements, and different approaches focus on reducing the cost of different components. For that reason, it is helpful to start by providing a breakdown of the computational cost of Transformer components. Table 1 shows the FLOPs[1] for each component of a Transformer encoder layer (Kaplan et al., 2020).

| Encoder Layer Component | Flops |
|---|---|
| Vanilla self-attention computation | $2n^2d$ |
| Attention QKV and output projections | $4nd^2$ |
| Feedforward layer | $8nd^2$ |
| LONGT5 local attention computation | $2nwd$ |
| LONGT5 global attention computation | $\frac{n^2}{8}d$ |

Table 1: Computational cost of encoder layer transformer components measured in FLOPs. $n$ is the input length, $d$ is the model dimensionality, and $w$ is the size of the local attention window.

**Sparse attention**   The first challenge of applying a Transformer to a long input is that the FLOPs of the self-attention mechanism scales quadratically in the input length, becoming intractable for long inputs. A large body of work focuses on reducing self-attention cost, restricting attention between a subset of inputs (Child et al., 2019; Ainslie et al., 2020; Beltagy et al., 2020; Zaheer et al., 2020; Wang et al., 2020; Guo et al., 2022) or to a subset of layers (Zemlyanskiy et al., 2021). In LONGT5 (Guo et al., 2022), the most closely related model to COLT5, tokens attend within a lo-

---

[1]Each multiply-add is counted as a single FLOP.

cal window as well as to a mean-pooled summary representation for each block of 16 tokens in the input. LONGT5 attention leads to sharply reduced (though still non-negligible) FLOPs (Table 1).

**Conditional computation** After applying a sparse attention mechanism, the feedforward and attention projection layers account for the majority of the FLOPs. These costs scale with the length of the input, such that processing long inputs is still prohibitively expensive. A common approach to reduce the remaining cost is to employ some form of *conditional computation*, avoiding applying all model parameters to the entire input. CALM (Schuster et al., 2022) applies a varying number of decoder layers to each decoded token, outputting a token early if the model is confident in its prediction. Mixture-of-Experts models (Shazeer et al., 2017; Fedus et al., 2021; Zoph et al., 2022) route inputs through a small proportion of expert sub-modules, bringing to bear only the parameters most relevant to the input. In the context of retrieval-augmented models, numerous works re-rank retrieved passages by their relevance to the query and process only the highest scoring passages (Mao et al., 2021; Wang et al., 2018; Yu et al., 2022) and vary the number of processed passages depending on model confidence (Kratzwald and Feuerriegel, 2018; Varshney et al., 2022). Concurrent work CoDA (Lei et al., 2023) employs a related conditional computation mechanism, designed for efficient adaptation rather than modeling long documents.

**Device utilization** FLOPs do not tell the whole story, as modeling choices can influence the effective speed of operations achieved by accelerators. For long text inputs, autoregressive decoder inference is very slow due to memory bandwidth constraints from repeatedly loading the long sequence of keys and values (Shazeer, 2019; de Jong et al., 2022). Shazeer (2019) introduces multi-query attention (MQA), sharing heads for keys and values to reduce memory bandwidth overhead. Pope et al. (2022) studies how to shard large models, especially in the context of MQA, to obtain optimal device utilization and therefore speed.

**Training objectives** T5 introduced the span corruption objective (Raffel et al., 2020), a modification of masked language modeling (Devlin et al., 2019). LONGT5 made use of the PEGASUS (Zhang et al., 2020) sentence reconstruc-

tion objective for improved summarization performance. Tay et al. (2022) proposes UL2, a mixture of span corruption, prefix, and causal language modeling, and shows that it leads to strong performance on both short-output and generative tasks.

# 3 CoLT5

## 3.1 Conditional computation

As discussed in the previous section, a large proportion of Transformer FLOPs arise from feedforward and projection layers that scale with the length of the input sequence. Therefore, LONGT5 training and inference on long documents remains expensive.

CoLT5 further reduces the cost of processing long documents through *conditional computation*, following the intuition that some tokens are more important and therefore benefit more than others from heavy computation. First, some types of tokens may inherently require less computation, such as filler words and punctuation. Second, especially in long documents, large parts of the input may not be relevant to the current question, task, or processing stage.

The CoLT5 conditional computation mechanism consists of three components: routing modules, conditional feedforward layers, and conditional attention layers. All tokens are processed by standard, lightweight attention and feedforward layers. Routing modules additionally select important tokens from an input at each attention or feedforward layer, and a heavy conditional layer applies additional computation to routed tokens. This section describes each component in detail. Figure 1 provides an overview of the CoLT5 conditional computation mechanism, and Table 2 compares CoLT5 and LONGT5 FLOPs.

| Model | Encoder Layer Flops |
|---|---|
| T5 | $12nd^2 + 2n^2d$ |
| LONGT5 | $12nd^2 + \frac{n^2}{8}d$ |
| CoLT5 | $7\frac{1}{4}nd^2 + \frac{n^2}{84}d$ |

Table 2: **CoLT5 uses significantly fewer FLOPs than LONGT5.** Comparison of approximate encoder layer total FLOPs between T5, LONGT5, and CoLT5. CoLT5 FLOPs rounded to readable fractions.

**Routing** In order to separately select important tokens for each component in each layer, we need

a *learnable* and *tractable* routing function. We follow the simple three-step mechanism from Lei et al. (2023): (1) multiply inputs with a learned embedding to obtain routing scores, (2) normalize, and (3) select the top-$k$ highest scoring inputs.

Let $X_i$ be the representation of token $i$, and $u$ a $d$-dimensional learnable embedding. Then the routing score of token $i$ is

$$s_i = X_i \cdot u$$

We select the top-$k$ highest scoring inputs. In order to provide a learning signal to the scoring embedding, we make sure the contribution of the routed tokens to the layer update is *scaled* according to the routing score, as will be seen later. To provide a better distributed signal to all tokens, we also globally normalize the routing scores to sum up to the number of desired routed tokens using a generalized softmax, resulting in normalized scores $\tilde{s}_i$. Each COLT5 layer has three independent routers, one each for the feedforward layer, attention queries, and attention key-values.

**Conditional Feedforward** Intuitively, some token representations may benefit from more processing than others. The COLT5 conditional feedforward layer applies an additional high-capacity feedforward layer to selected tokens. In particular, let $X_i$ be the model state of the $i$th token and $\tilde{s}_i$ denote the normalized routing score (set to 0 for non-routed tokens). Then the feedforward update for COLT5 is given by

$$X_i = X_i + \text{FFd}_{\text{Light}}(X_i) + \tilde{s}_i \cdot \text{FFd}_{\text{Heavy}}(X_i)$$

The light and heavy feedforward branches differ only in their hidden dimension, with the light branch having smaller hidden dimension than the standard T5 feedforward layer and the heavy branch larger. Let $n$ denote the number of input tokens, $m$ the number of selected tokens, and $r_L$ and $r_H$ the ratios of light and heavy hidden dimension to standard T5 hidden dimension. Then the FLOPs of the COLT5 layer are given by

$$\text{FLOPs}_{\text{FFd}} = \underbrace{8nr_Ld^2}_{\text{Light branch}} + \underbrace{8mr_Hd^2}_{\text{Heavy branch}}$$

We set the light and heavy ratios as $r_L = \frac{1}{2}$ and $r_H = 4$, half and quadruple the standard T5 hidden dimension respectively. For our main experiments, a fraction $\frac{1}{16}$ of tokens are routed to the

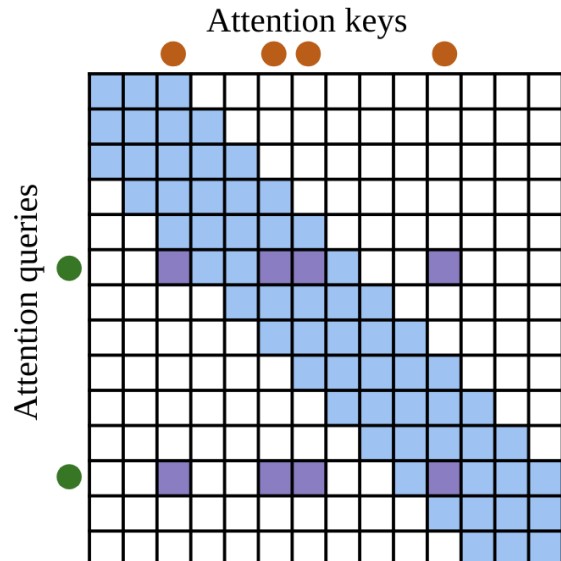

Figure 3: An overview of the COLT5 attention pattern. The light branch performs local attention for each token. In the higher capacity heavy branch $q$ selected query tokens (2 in the figure) attend to $v$ separately selected key and value tokens (4 in the figure).

heavy branch. As a result the approximate FLOPs from the COLT5 feedforward layer equals

$$\text{FLOPs}_{\text{FFd}} = \underbrace{4nd^2}_{\text{Light branch}} + \underbrace{2nd^2}_{\text{Heavy branch}}$$

consuming 75% of the FLOPs of a standard T5 feedforward layer.

**Conditional Attention** COLT5 conditional attention operates on the intuition that most tokens have simple, local interactions, but some tokens benefit from heavier processing and long-range interactions. The COLT5 conditional attention layer applies an additional high-capacity attention layer that attends from selected query tokens to selected key-value tokens. Let $\tilde{s}_i^q$ denote the normalized routing query score for token $i$, and $\tilde{s}^{kv}$ the key-value scores for all tokens (set to 0 if not routed). Then the attention update for COLT5 is given by

$$X_i = X_i + \text{A}_{\text{Light}}(X_i, X) + \tilde{s}_i^q \cdot \text{A}_{\text{Heavy}}(X_i, \tilde{s}^{kv}X)$$

The light and heavy branches differ in the number of heads and tokens attended to: the light branch has fewer heads and attends to a local context window, while the heavy branch has more heads and attends to all routed key-value tokens. Separately selecting query and key-value tokens also allows the model to differentiate between tokens that *require* additional information and those that *possess*

| Model | Avg | Speed | | TQA | NQA | QAS | QuAL | CNLI | arXiv | SumS | QMS | GovR |
|---|---|---|---|---|---|---|---|---|---|---|---|---|
| | | inf | fn | F1 | F1 | F1 | EM | EM | $R_{gm}$ | $R_{gm}$ | $R_{gm}$ | $R_{gm}$ |
| LONGT5-B | 43.1 | 0.6 / 7.4 | 3.7 | 82.2 | 23.0 | 46.6 | 37.9 | 85.6 | 35.4 | 19.2 | 20.4 | 37.7 |
| COLT5-B | 42.4 | 11.2 | 6.5 | 82.4 | 23.3 | 42.1 | 36.5 | 86.5 | 35.3 | 18.7 | 18.4 | 37.9 |
| LONGT5-L | 45.3 | 0.3 / 3.0 | 1.3 | 84.2 | 27.2 | 52.3 | 40.6 | 87.3 | 35.7 | 19.1 | 21.4 | 39.5 |
| COLT5-L | 45.3 | 5.0 | 2.0 | 84.5 | 27.7 | 49.8 | 39.9 | **88.7** | 35.9 | **20.5** | 21.0 | 39.7 |
| LONGT5-XL | 46.6 | 0.2 / 1.2 | 0.4 | 85.3 | 29.3 | 53.1 | 46.0 | 88.2 | 35.9 | 19.4 | 21.3 | **40.5** |
| COLT5-XL | **47.4** | 2.3 | 0.5 | **86.1** | **31.1** | **53.9** | **48.1** | 88.4 | **36.1** | 20.0 | **22.5** | 40.5 |

Table 3: Performance comparison of COLT5 and LONGT5 Base, Large and XL models on question-answering datasets TriviaQA (TQA), NarrativeQA (NQA), QASPER (QAS), and QuALITY (QuAL), NLI dataset ContractNLI (CNLI), and summarization datasets arXiv, SummScreenFD (SumS), QMSum (QMS), and GovReport (GovR). SCROLLS results are on leaderboard test set where COLT5-XL achieves SOTA. Average speed is reported in samples per second for inference (inf) and fine-tuning (fn). LONGT5 does not use MQA but inference speed is reported without/with MQA for conservative baseline. $R_{gm}$ stands for the geometric mean of ROUGE-1,2,L. Similar to SCROLLS, we take a simple average across all datasets even though the datasets use different performance metrics.

such information. Figure 3 shows the COLT5 attention pattern. Let $q, v$ be the number of selected query and key-value tokens, $w$ the size of the local attention window and $r_L, r_H$ the proportion of light and heavy heads relative to standard T5. Then the FLOPs of the COLT5 attention layer are given by

$$\text{FLOPs}_{\text{Att}} = \underbrace{4n \cdot r_L d^2}_{\text{Local projection}} + \underbrace{2nw \cdot r_L d}_{\text{Local attention}}$$
$$+ \underbrace{2q \cdot r_H d^2 + 2v \cdot r_H d^2}_{\text{Global projection}} + \underbrace{2qv \cdot r_H d}_{\text{Global attention}}$$

We set the light and heavy head ratios as $r_L = \frac{1}{4}$ and $r_H = \frac{3}{4}$, keeping the total number of heads across the light and heavy branches equal to standard T5 heads. For our main experiments a fraction $\frac{1}{16}$ query tokens and $\frac{1}{8}$ key-value tokens are routed to the heavy branch, so $q = \frac{n}{16}$ and $v = \frac{n}{8}$. Ignoring local attention computation, we approximate attention FLOPS by[2]

$$\text{FLOPs}_{\text{Att}} \approx \underbrace{nd^2}_{\text{Local proj.}} + \underbrace{\frac{1}{4}nd^2}_{\text{Global proj.}} + \underbrace{\frac{1}{84}n^2 d}_{\text{Global att.}}$$

with less than half projection FLOPs and order-of-magnitude smaller quadratic length scaling compared to LONGT5. Table 2 shows total FLOPs for the COLT5 layer. In general, we set $q = m$ and $v = 2m$, and use $m$ to summarize the number of routed tokens going forward.

---

[2]Global projection and attention FLOPs rounded to readable fractions, exact values are $\frac{9}{32}$ and $\frac{3}{256}$. Complexity assumes constant fraction of routed tokens; we show we can do better in practice for extremely long inputs.

### 3.2 Multi-query Attention

Conditional computation effectively reduces the computational cost of the encoder. However, for encoder-decoder models with long inputs the majority of inference time is spent in the decoder due to memory bandwidth constraints (Shazeer, 2019; de Jong et al., 2022). Most of the overhead is caused by repeatedly reading all the input token keys and values from memory for every output token that is autoregressively decoded during cross attention. Multi-query attention (Shazeer, 2019) (MQA) allows all query heads to share a single key and value head, alleviating this bottleneck. Accordingly, we apply MQA in cross-attention layers for much faster inference. Note however that MQA does not improve training speed since target tokens are processed in parallel during training, avoiding this memory bandwidth bottleneck.

### 3.3 UL2

The UL2 pre-training objective (Tay et al., 2022) combines different denoising objectives, extending the span corruption pre-training used in T5 to a variety of noise rates / average span lengths and adding a prefix language modeling objective more similar to typical decoder-only model pre-training. UL2 has been shown to lead to improved in-context learning. We train COLT5 on UL2 instead of PEGASUS (Zhang et al., 2020), endowing COLT5 with in-context learning capabilities.

## 4 Experiments

In order to evaluate COLT5, we perform the following experiments: (1) our main results com-

pare CoLT5 and LongT5 on a collection of long input datasets using input length of 16k tokens; (2) we evaluate CoLT5 on extremely long inputs up to 64k tokens and compare scaling against LongT5; (3) demonstrate CoLT5's few-shot capability, investigating how performance changes as input length and number of shots increase, (4) perform a series of ablations to understand the effect of individual CoLT5 components, and (5) investigate empirical routing patterns. The remainder of the section outlines our experimental setup, and then describes each of the experiments above.

## 4.1 Experimental setup

**Configurations** CoLT5 is based on the T5.1.1 architecture (Raffel et al., 2020), implemented with JAX (Bradbury et al., 2018), Flax (Heek et al., 2020), and Flaxformer[3]. Following LongT5, we experiment with Base, Large, and XL model sizes. CoLT5 models use the same embedding dimension, number of layers, and total attention heads as corresponding LongT5 models of the same size, with more overall parameters (but less compute) due to the conditional branch. See Appendix B for additional details on model configuration.

**Pre-training** We pre-train CoLT5 for 1M steps on the C4 dataset (Raffel et al., 2020) using a variant of the UL2 objective (Tay et al., 2022) with batch size 256, input length 4096, and output length 910. In particular, our mixture contains four objectives in equal proportion: prefix-LM with noise rate 0.5, and span corruption (Raffel et al., 2020) with noise rate 0.15 and average span lengths 3, 8, and 64. We use the Adafactor optimizer (Shazeer and Stern, 2018) with the T5.1.1 inverse square root learning rate schedule and no dropout. CoLT5 is trained with the T5X (Roberts et al., 2022) framework. For pre-training, we route $m = 512$ tokens, $\frac{1}{8}$th of the input length.

**Fine-tuning** For fine-tuning we use a constant learning rate of 0.001, batch size 128, and dropout rate 0.1 for all tasks. Main results use input length of 16384 for all datasets other than ContractNLI, which uses 8192. Question answering datasets use output length 128 and summarization datasets use output length 512, except for GovRep which uses output length 1024. We route $m = 1024$ tokens, $\frac{1}{16}$th of the input length. We train until convergence

---

[3]https://github.com/google/flaxformer

and select the checkpoint with the highest dev performance. We use greedy decoding for inference.

**Data** We evaluate CoLT5 on TriviaQA (Joshi et al., 2017), arXiv (Cohan et al., 2018), and the SCROLLS benchmark (Shaham et al., 2022). SCROLLS contains question-answering datasets: NarrativeQA (Kočiský et al., 2018), QASPER (Dasigi et al., 2021), and QuALITY (Pang et al., 2021), an NLI dataset: ContractNLI (Koreeda and Manning, 2021), and summarization datasets: SummScreenFD (Chen et al., 2022), QMSum (Zhong et al., 2021), and GovReport (Huang et al., 2021). Table 4 provides an overview of the size and input length for each dataset.

| Dataset | Type | Samples | Median | 90% |
|---|---|---|---|---|
| TriviaQA | QA | 157,053 | 8,858 | 28,956 |
| arXiv | Sum | 215,913 | 8,519 | 20,170 |
| NarrativeQA | QA | 71,187 | 57,829 | 176,862 |
| QASPER | QA | 5,692 | 5,472 | 8,657 |
| QuALITY | QA | 6,737 | 7,171 | 8,276 |
| ContractNLI | NLI | 10,319 | 2,148 | 4,485 |
| SummScreen | Sum | 4,348 | 9,046 | 15,172 |
| QMSum | Sum | 1,810 | 14,197 | 27,761 |
| GovRep | Sum | 19,402 | 8,841 | 18,835 |

Table 4: Median and 90th percentile input length by dataset measured in SentencePiece tokens.

**Timing** We report time per sample per TPUv4 chip, as measured by xprof (Google, 2020). For inference we use a single TPUv4 with batch size 16 or the largest that fits in memory. For fine-tuning we profile with 8 TPUv4 chips, sharded separately for each model to maximize throughput.

## 4.2 Main results

Figure 2 compares the quality-speed trade-off for LongT5[4] and CoLT5, showing that CoLT5 is better at any speed. For 16k input length, CoLT5 matches or exceeds LongT5 quality for Large and XL with 35-75% training speedup and 50-100% inference speedup on top of the order-of-magnitude inference speedup from MQA. Encoder speedups are even greater (Appendix D). CoLT5-XL also achieves SOTA performance on the SCROLLS benchmark. Table 3 contains all main results.

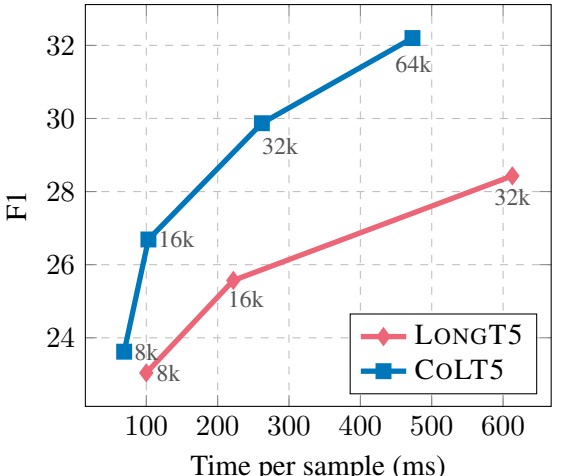

Figure 4: **COLT5 effectively scales to extremely long inputs, achieving stronger performance and faster speed than LONGT5.** F1 on NarrativeQA as a function of inference time per sample for LONGT5 and COLT5 Large models using varying input lengths.

## 4.3 Scaling to extremely long inputs

We hypothesize that the advantage of COLT5 over LONGT5 strengthens with input length, as the fraction of important tokens decreases and COLT5 can route a greater proportion of important tokens to the heavy branch. Figure 4 compares the quality-speed trade-off for LONGT5 and COLT5 on NarrativeQA, sweeping over input length rather than model size. The number of routed tokens is $\frac{1}{16}$th of the input length, except that we do not increase routed tokens going from 32k to 64k, so at 64k we route only $\frac{1}{32}$nd of the input length. COLT5 achieves both stronger performance and faster inference speed at all input lengths and is able to effectively make use of extremely long inputs. We note that COLT5 achieves large quality gains by going from 32k to 64k tokens even while keeping the number of routed tokens constant, providing more evidence for our hypothesis.

## 4.4 In-context learning

Models trained on the UL2 objective have shown strong few-shot in-context learning (ICL) capabilities[5] even at smaller sizes (Tay et al., 2022). COLT5 enables tractable inference with long inputs. Here, we leverage this for scaling the number of examples used for in-context learning.

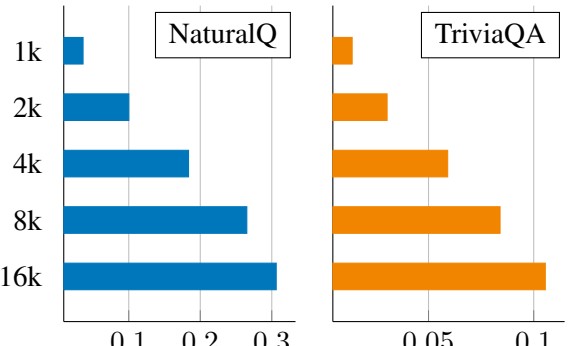

Figure 5: **COLT5 can use its long-input capability to benefit from more shots for in-context learning.** Few-shot exact match for COLT5-Large on Natural Questions and TriviaQA dev sets as a function of input tokens, fitting as many examples as possible. Each example contains question, context, and answer. Inputs length used are 1024, 2048, 4096, 8192, 16384.

We test the above hypothesis by evaluating few-shot learning performance on Natural Questions (Kwiatkowski et al., 2019) and TriviaQA as a function of input length, using as many examples as fit in the context. We consider the open book setting, such that each example consists of question, context document, and answer. Table 5 shows the number of examples by input length. We evaluate on the full dev set, randomly sampling examples from the training set for each dev sample until no further examples fit in the input length. We found that COLT5 can perform in-context learning only up to the input length it was trained on, so for these experiments we continued pre-training a COLT5-Large model on input length 16384 for another 100k steps. For the same reason we route $m = 512$ tokens as in pre-training.

Figure 5 displays COLT5 few-shot performance as a function of input length, showing that COLT5 is able to apply its long-input capabilities to extract information from increasing numbers of examples.

| Dataset | 1024 | 2048 | 4096 | 8192 | 16384 |
|---------|------|------|------|------|-------|
| NQ | 0.1 | 0.7 | 1.7 | 3.4 | 5.6 |
| TriviaQA | 1.6 | 2.3 | 3.8 | 7.0 | 9.8 |

Table 5: Average number of Natural Questions and TriviaQA few-shot examples that fit in input length.

## 4.5 Ablations

This section studies the effect of different choices in the COLT5 recipe. Table 6 contains results of a series of experiments that change a single compo-

---

[4]Note that LONGT5 does not use MQA, but for profiling we add MQA to LONGT5 for a conservative baseline.

[5]We initially evaluated ICL for models pre-trained with PEGASUS but found performance to be nearly 0.

| Ablation | Model | Avg | Inf | TQA | NQA | QAS | QuAL | CNLI | arX | SumS | QMS | GovR |
|---|---|---|---|---|---|---|---|---|---|---|---|---|
| | | | S/s | F1 | F1 | F1 | EM | EM | R$_{gm}$ | R$_{gm}$ | R$_{gm}$ | R$_{gm}$ |
| Baseline | COLT5-B | 42.5 | 11.2 | 82.4 | 23.1 | 38.3 | 36.6 | 87.8 | 35.3 | 19.3 | 20.5 | 39.4 |
| Routing | Static | 40.5 | 11.6 | 79.7 | 19.2 | 34.2 | 34.5 | 86.4 | 34.9 | 18.1 | 18.9 | 38.8 |
| | Share QKV | 42.0 | 11.8 | 82.1 | 21.9 | 37.5 | 36.2 | 87.0 | 35.2 | 18.2 | 20.4 | 39.7 |
| Attention | v=all | 42.5 | 9.4 | 82.4 | 22.3 | 38.6 | 37.2 | 87.8 | 35.3 | 19.1 | 20.3 | 39.8 |
| | v=q | 42.3 | 11.5 | 82.5 | 22.5 | 37.3 | 37.0 | 85.9 | 35.2 | 19.0 | 20.5 | 39.7 |
| Routed Tokens | m=512 | 41.6 | **12.2** | 81.9 | 22.1 | 37.3 | 35.4 | 84.6 | 35.2 | 18.9 | 19.5 | 39.6 |
| | m=1536 | **42.9** | 10.4 | 82.6 | **23.5** | 39.8 | **37.5** | 87.5 | 35.4 | 19.4 | 20.8 | 40.0 |
| Encoder | LONGT5-B | 42.1 | 7.4 | 82.0 | 21.4 | 38.4 | 35.8 | **88.0** | 35.5 | 18.7 | 20.4 | 38.5 |
| Decoder | Multi-head | **42.9** | 0.7 | **82.7** | 22.9 | 40.2 | 35.8 | 87.7 | 35.5 | **19.7** | **21.2** | **40.3** |
| Objective | PEGASUS | 42.8 | 11.2 | 82.6 | 22.6 | **40.5** | 37.3 | 87.3 | 35.3 | 19.6 | 20.8 | 39.6 |

Table 6: COLT5 ablations evaluated on validation sets. Each experiment modifies a component of the COLT5 recipe for COLT5-Base. Static routing divides the input into equal-length blocks and selects the first token in each block to be routed. Shared QKV routing shares routing decisions for queries and keys/values. In v=all the routed queries attend to the entire input, while v=q selects the same number of key and value tokens as query tokens. m=512 and m=1536 use different numbers of routed tokens. LONGT5-B uses a LONGT5 encoder while retaining other parts of the COLT5 training recipe such as MQA and the UL2 objective. Multi-head refers to using multi-head cross-attention. The final ablation replaces the UL2 objective with PEGASUS as in LONGT5.

nent for COLT5 Base.

**Routing** First, we note that static routing -- evenly distributing routed tokens over the input -- leads to massive drop in performance. The importance of routing provides evidence that the model learns to devote capacity to important tokens and the advantage of COLT5 is not merely a result of additional parameters. Sharing routing decisions for query and KV tokens should be compared with v=q, and leads to a modest reduction in quality and increase in speed.

The optimal number of routed tokens represents a trade-off between improved performance and computational cost of applying heavier layers. Table 6 shows strong gains going from 512 to 1024 (baseline) routed tokens and diminishing returns for further increases.

**Attention** COLT5 relies on routing to identify not only tokens that can benefit from important information elsewhere in the input, but also which tokens contain such important information. We study whether COLT5 is successful in this task by comparing performance with two different attention settings -- v=all, in which routed tokens attend to the entire input, and v=q, which uses equal number of routed keys and values as queries, rather than twice as many. COLT5 appears to occupy a sweet spot, as using fewer routed key-values modestly decreases performance at similar speed but attending to all inputs barely helps at sharply increased cost.

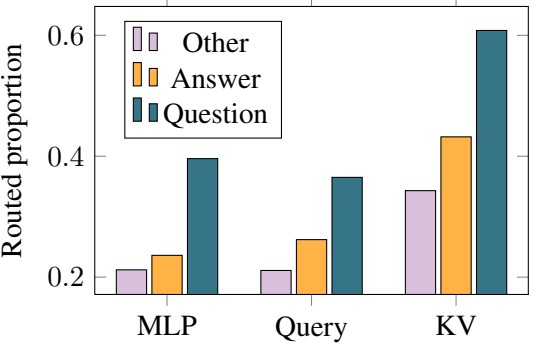

Figure 6: Proportion of tokens routed for answer (string match), question, and other tokens by routing component for COLT5 Large model, averaged over examples in TriviaQA dev set and all layers of model.

**Other** We compare COLT5 to LONGT5 with multi-query cross-attention, confirming that LONGT5 indeed does not achieve an unexpected quality gain from MQA, and our conservative assumptions in Figures 2, 4 are valid. Next, we evaluate multi-head cross-attention for COLT5, finding that it leads to modestly improved COLT5 performance. However, as MHA exhibits order-of-magnitude slower inference, MQA is clearly favored. Finally, PEGASUS appears to fine-tune slightly better than UL2, though the difference is small and UL2 enables few-shot learning.

## 4.6 Routing analysis

It is interesting to ask whether COLT5 routed tokens line up with what we consider intuitively important tokens in each document. We investigate this question by studying routing patterns of a Large COLT5 model fine-tuned on TriviaQA. We divide tokens into three categories: (1) question tokens, (2) answer tokens, and (3) other tokens. Figure 6 shows the average fraction of each type of token that is routed through the heavy path for MLP and attention layers on TriviaQA. We note that question and answer tokens are significantly more likely to be routed than other tokens, for feedforward as well as attention queries and keys/values. Appendix F presents more detailed routing analysis; e.g., semantically important tokens are much more likely to be selected in later layers.

## 5 Conclusion

We propose COLT5, a new model for long-range inputs that employs conditional computation for higher quality and faster speed. COLT5 has light feedforward and attention layers that apply to the entire input, as well as heavy branches that are applied only to a subset of important tokens selected by a learned router. We show that COLT5 achieves stronger performance at any speed compared to LONGT5 on a variety of long-input datasets, and can effectively and efficiently make use of extremely long inputs up to 64k tokens.

## Limitations

COLT5 applies conditional computation only in the encoder. Applying conditional computation in the decoder is more complicated; the routing method in COLT5 is not causal, so it isn't applicable when generating token by token. Since decoder-only models and applications with long outputs have become more popular recently, this is a strong limitation of the current approach. Although the routing method in COLT5 could potentially be applied to the *input* context in a decoder-only model, we didn't investigate this setup.

COLT5 is specialized towards long sequences and has to be trained from scratch. For large-scale training and deployment, it is desirable to either train a single model that can handle both short and long sequences, or develop a long-input architecture that can be adapted from an existing large model.

## Acknowledgements

We would like to thank Srinadh Bhojanapalli, Luke Vilnis, Zachary Fisher, Jianmo Ni, Tal Schuster, Vaclav Cvicek, Sudeep Gandhe, Bhargav Kanagal, Kenton Lee, Ming-Wei Chang, Afroz Mohiuddin, Raphael Hoffmann, and others at Google Research for helpful advice and discussion.

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

| Model | Layers | Model dim | MLP$_{light}$ dim | MLP$_{heavy}$ dim | Heads$_{light}$ | Heads$_{heavy}$ | Params |
|---|---|---|---|---|---|---|---|
| LONGT5-B | 12 | 768 | 2048 | N/A | 12 | N/A | 248m |
| COLT5-B | 12 | 768 | 1024 | 8096 | 4 | 8 | 433m |
| LONGT5-L | 24 | 1024 | 2816 | N/A | 16 | N/A | 783m |
| COLT5-L | 24 | 1024 | 1408 | 11264 | 4 | 12 | 1462m |
| LONGT5-XL | 24 | 2048 | 5120 | N/A | 32 | N/A | 2850m |
| COLT5-XL | 24 | 2048 | 2560 | 20480 | 8 | 24 | 5297m |

Table 7: Hyperparameters for LONGT5 and COLT5 models. T5.1.1 hyperparameters match LONGT5. COLT5 parameters are sparsely accessed as a result of conditional computation, so parameter counts do not reflect compute, and for a given model size COLT5 is in fact faster than LONGT5 despite having more parameters.

## A    Contributions

Joshua led the project, developed the initial conditional attention mechanisms, and conducted most experimental ablations. Tao developed the heavy/light formulation for heterogeneous conditional computation, comprising the routing and conditional feedforward mechanisms, and iterated with Joshua on initial experiments demonstrating feasibility. Michiel helped to scope the paper, performed most of the writing, and oversaw speed benchmarking. Santiago designed and conducted all the few-shot experiments, initiated the routing analysis visualization, and integrated UL2 into the codebase. Siddhartha developed the separate routing for query and key/value tokens in the conditional attention component and demonstrated the resulting quality improvements. Yury designed and conducted all experiments for inputs larger than 16k tokens, demonstrating favorable scaling up to 64k. David integrated all SCROLLS tasks into the codebase and ran early experiments, especially comparing UL2 with PEGASUS. Mandy developed the leaderboard comparisons with LongT5 and helped run several experiments. James advised on and ran early comparisons with MoE conditional

computation. Yi advised on the adaptation of UL2 to 4k input length pre-training. Finally, Yun-Hsuan and Sumit provided guidance and support for the project overall.

## B    Model Hyperparameters

Table 7 shows LONGT5 and COLT5 hyperparameters, including parameter counts. For LONGT5, we report numbers for the TGlobal configuration, which match T5.1.1. Notice that COLT5's parameter counts are larger due to using conditional compute. Similar to other conditional compute architectures such as mixture-of-experts, computational cost does not necessarily increase with parameter count.

We use the same 127-token local radius for COLT5 as LONGT5. This results in a local attention window $w$ of 255 since 127 tokens are attended to the left and 127 to the right.

## C    Routing Normalization Hyperparameters

To normalize the routing scores for differentiable top-$k$ token selection, we use the iterative soft top-$k$ algorithm from Lei et al. (2023) and Qian et al.

| Model | Average | | 16k in, 128 out | | 16k in, 512 out | | 16k in, 1024 out | | 8k in, 128 out | |
|---|---|---|---|---|---|---|---|---|---|---|
| | Enc | Tot | Enc | Tot | Enc | Tot | Enc | Tot | Enc | Tot |
| LONGT5-B | 77 | 136 | 84 | 98 | 84 | 165 | 84 | 296 | 27 | 39 |
| COLT5-B | 29 | 90 | 30 | 45 | 30 | 113 | 30 | 256 | 18 | 30 |
| LONGT5-L | 164 | 329 | 173 | 222 | 179 | 392 | 179 | 799 | 66 | 100 |
| COLT5-L | 70 | 201 | 73 | 103 | 73 | 250 | 73 | 578 | 45 | 69 |
| LONGT5-XL | 390 | 870 | 412 | 557 | 423 | 1081 | 423 | 2065 | 166 | 290 |
| COLT5-XL | 177 | 439 | 185 | 239 | 185 | 525 | 185 | 1253 | 115 | 163 |

Table 8: Comparison of total and encoder inference time per sample (ms) for LONGT5 and COLT5 Base, Large, and XL models at different input and output lengths. Average time per sample is computed as a weighted average over input and output lengths, weighted by the number of tasks in our evaluation that use the corresponding setting (4 for 16k/128, 3 for 16k/512, and one each for 16k/1024 and 8k/128).

| Model | arXiv | | | SummScreenFD | | | QMSum | | | GovRep | | |
|---|---|---|---|---|---|---|---|---|---|---|---|---|
| | R-1 | R-2 | R-L | R-1 | R-2 | R-L | R-1 | R-2 | R-L | R-1 | R-2 | R-L |
| LONGT5-B | 47.4 | 21.4 | 43.5 | 34.8 | 9.3 | 20.7 | 35.1 | 11.1 | 23.4 | 59.3 | 30.1 | 33.0 |
| COLT5-B | 47.5 | 21.3 | 43.6 | 35.6 | 9.7 | 21.0 | 34.6 | 10.9 | 23.0 | 60.2 | 31.0 | 32.8 |
| LONGT5-L | 47.9 | 21.7 | 43.8 | 35.3 | 9.1 | 20.8 | 35.9 | 12.0 | 24.1 | 61.4 | 32.5 | 34.1 |
| COLT5-L | 48.4 | 21.7 | 44.3 | 35.7 | 10.1 | 21.4 | 36.8 | 12.6 | 24.7 | 61.8 | 32.7 | 34.4 |
| LONGT5-XL | 48.2 | 21.8 | 44.1 | 36.6 | 10.3 | 21.5 | 37.0 | 12.5 | 24.7 | 61.8 | 33.2 | 34.8 |
| COLT5-XL | 48.4 | 22.0 | 44.3 | 36.3 | 10.0 | 21.5 | 37.4 | 13.0 | 25.1 | 62.2 | 33.3 | 34.9 |

Table 9: Full performance comparison with Rouge-1, Rouge-2, and Rouge-L metrics of COLT5 and LONGT5 Base, Large, and XL models on summarization dev sets. Results based on checkpoint that maximizes $R_{gm}$ as in Table 3.

(2022) with $\epsilon = 1.0$ and 50 iterations. During training we allow the top $\frac{9}{8}k$ tokens to have nonzero weight instead of just the top $k$ in order to provide a slightly improved training signal.

## D Additional Experimental Results

Table 8 compares LONGT5 and COLT5 inference speed in more detail, splitting off encoder and total time per sample. Since COLT5 applies conditional computation only in the encoder, encoder speed gains are larger than overall speed gain, and total speed gains are largest for shorter output length. Trade-offs are even more in the favor of COLT5 when paired with other decoder optimizations.

Table 9 shows full (Rouge-1, Rouge-2, Rouge-L) results for summarization datasets.

## E Computational Resources

For pre-training we generally used 128 TPUv4 chips for Base and 256 TPUv4 chips for Large and XL. Pre-training took approximately 2.5 days for Base, 3.7 days for Large, and 12.8 days for XL. For fine-tuning we generally used 64, 128, and 256 TPUv4 chips for Base, Large, and XL, respectively, with training time varying with dataset size.

## F Routing Analysis

In this section we take a closer look at the routing mechanisms in COLT5. There are three routing processes in each layer of COLT5: (1) Routing of attention keys and values ("KV-routing"), (2) routing of attention queries ("Q-routing") and (3) routing of MLP tokens ("MLP-routing"). For simplicity, we will say that a token is *selected*, when it is routed to the heavy alternative (of either MLP or attention). We are interested in understanding what tokens are selected and whether these mechanisms select similar or different tokens in each layer.

**Which tokens are selected** We divide input tokens into three categories: (1) question tokens, (2) answer tokens (found via simple normalized string match of the ground truth answer), and (3) other tokens. Figure 7 shows the proportion of each token type that is routed by a fine-tuned COLT5-Large model on the TriviaQA dev set, by layer and routing component.

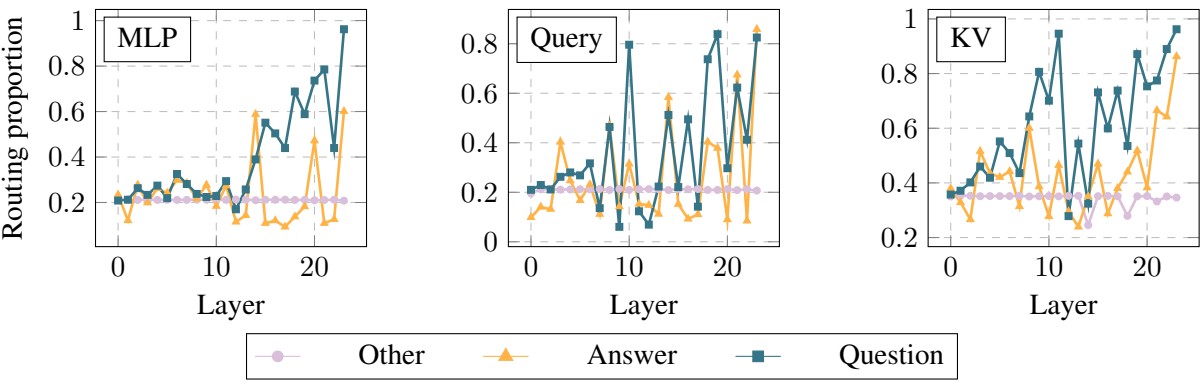

Figure 7: Proportion of tokens routed for answer (string match), question, and other tokens by routing component and layer for COLT5 Large model, averaged over examples in TriviaQA dev set.

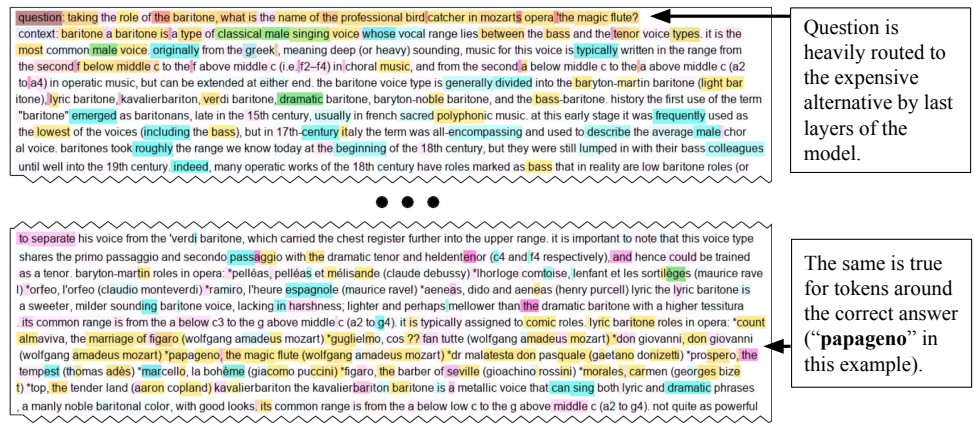

Figure 8: Visualization of token routing weights for some fragments of an example on TriviaQA.

Earlier we showed that question and answer tokens are more likely to be selected, but separating routing decisions by layer reveals interesting patterns. At early layers question and answer tokens are only modestly more likely to be selected, with routing probability sharply increasing at later layers and peaking in the last layer. This makes intuitive sense: in early layers the model has not yet had the opportunity to identify which tokens and parts of the document are important. However, the increase is not monotonic and there is strong variation between layers. This variation may imply that different layers focus on different types of tokens, or that some routing components do not successfully learn to identify important tokens.

To gain a better insight into this, Figure 8 visualizes routing on two sample fragments from a TriviaQA example (notice that, given the large input length used in CoLT5, we do not show the complete example in the figure). The two fragments shown correspond to the beginning of the example (where the question is located), and the part of the context surrounding the correct answer. We have added a colored background to the figure, where each of the three CMY channels are mapped to the KV-routing weights in different layers of the model. *Cyan* corresponds to layer 1, *Magenta* to layer 12, and *Yellow* to layer 24. As we can see, question and answer are heavily yellow colored, showing those tokens are selected in the last layer.

**Correlation between routing processes.** Table 10 shows the Pearson correlation coefficient between the routing weights of the different routing mechanisms in each layer in a CoLT5 *Large* model (MLP-routing correlation with KV-routing, MLP-

routing with Q-routing, and KV-routing with Q-routing). We show numbers for both the pre-trained checkpoint, as well as a fine-tuned model on TriviaQA. As we can see, the routing of keys/values and routing of queries is highly correlated at all layers except the first two, while the routing of tokens in the MLP has lower correlation to the other two processes. Interestingly correlation between MLP and attention routing increases in the last layers of the model.

|    | Pre-trained | | | Fine-tuned | | |
|----|--------|-------|------|--------|-------|------|
|    | **MLP-KV** | **MLP-Q** | **KV-Q** | **MLP-KV** | **MLP-Q** | **KV-Q** |
| 1  | -0.06 | -0.06 | -0.09 | -0.06 | -0.09 | -0.26 |
| 2  | 0.27  | 0.52  | 0.04  | 0.27  | 0.39  | 0.02  |
| 3  | -0.05 | -0.03 | 0.75  | 0.05  | -0.01 | 0.69  |
| 4  | 0.05  | 0.09  | 0.76  | 0.18  | 0.14  | 0.72  |
| 5  | 0.02  | -0.01 | 0.75  | 0.22  | 0.26  | 0.68  |
| 6  | 0.02  | -0.01 | 0.78  | 0.31  | 0.33  | 0.70  |
| 7  | 0.02  | 0.00  | 0.73  | 0.26  | 0.27  | 0.70  |
| 8  | 0.00  | -0.02 | 0.44  | 0.11  | -0.07 | 0.29  |
| 9  | 0.13  | 0.11  | 0.74  | 0.36  | 0.40  | 0.70  |
| 10 | -0.06 | -0.08 | 0.08  | -0.15 | -0.15 | 0.12  |
| 11 | -0.05 | -0.07 | 0.31  | -0.08 | -0.03 | 0.18  |
| 12 | -0.04 | -0.08 | 0.27  | 0.03  | 0.00  | 0.28  |
| 13 | -0.10 | -0.09 | 0.87  | -0.13 | -0.03 | 0.72  |
| 14 | -0.04 | -0.05 | 0.76  | -0.06 | -0.12 | 0.67  |
| 15 | 0.53  | 0.64  | 0.69  | 0.51  | 0.55  | 0.67  |
| 16 | 0.08  | 0.12  | 0.63  | 0.06  | 0.57  | 0.24  |
| 17 | 0.28  | 0.30  | 0.65  | 0.27  | 0.32  | 0.69  |
| 18 | 0.28  | 0.02  | 0.84  | 0.31  | 0.20  | 0.76  |
| 19 | 0.45  | 0.77  | 0.59  | 0.19  | 0.38  | 0.64  |
| 20 | 0.30  | 0.39  | 0.64  | 0.38  | 0.47  | 0.62  |
| 21 | 0.05  | -0.04 | 0.49  | 0.18  | 0.11  | 0.47  |
| 22 | 0.05  | 0.00  | 0.69  | 0.21  | 0.16  | 0.68  |
| 23 | 0.39  | 0.33  | 0.68  | 0.60  | 0.79  | 0.69  |
| 24 | 0.43  | 0.39  | 0.59  | 0.57  | 0.63  | 0.65  |

Table 10: Pearson correlation coefficient between the routing weights of the different routing mechanisms in each layer in a COLT5 *Large* model. We show numbers for both the pre-trained checkpoint, as well as a fine-tuned model on TriviaQA. Blue bars visualize positive correlation, whereas red bars visualize negative correlation.