# OpenReview forum: "CoLT5: Faster Long-Range Transformers with Conditional Computation"
_EMNLP/2023/Conference — EMNLP 2023 Main_

### Official Review · Reviewer_gBi6 · 2023-08-05

**Soundness:** 4

**Excitement:**

4: Strong: This paper deepens the understanding of some phenomenon or lowers the barriers to an existing research direction.

**Paper Topic And Main Contributions:**

The authors present a series of improvements to the T5 language model architecture, focusing on both increased input context and improvements to speed for both training and inference. Specifically, they extend the LongT5 model with a routing mechanism that examines only a portion of the input sequence deemed importance by a learnable score with a 'heavy' component with more network resources and a lower-dimensional 'light' component that attends to the entire input sequence. Multi-query cross-attention is used instead of standard multi-head attention to improve inference speed and the UL2  (span corruption, prefix, causal LM) training objective is used in place of LongT5's PEGASUS sentence reconstruction objective.

**Reasons To Accept:**

The authors carefully set up experiments to show how their approaches both improve on training and inference time and how each of the algorithmic improvements compares to the LongT5 model presented as the baseline. Training speed increases of 35-75% and inference speed increases of 50-100% are shown as well as improvements in output quality when increasing input token sizes from 16k to 64k.

**Reasons To Reject:**

The improvements to the model architecture do increase speed and performance by using computational 'tricks', but do not address overall algorithm complexity (e.g. the transformer algorithm used still has overall quadric complexity.) However, the authors architecture choices do have concrete runtime improvements when considering potential real-world use cases.

The authors state this type of model needs to be trained from scratch, which may be prohibitive for some compute-constrained research organizations. This can be mitigated by providing a pretrained base model to serve as a starting point.

**Reproducibility:**

4: Could mostly reproduce the results, but there may be some variation because of sample variance or minor variations in their interpretation of the protocol or method.

**Reviewer Confidence:**

4: Quite sure. I tried to check the important points carefully. It's unlikely, though conceivable, that I missed something that should affect my ratings.

**Typos Grammar Style And Presentation Improvements:**

It would be nice to see the "best" scores for each column highlighted in Table 6 (similar to Table 3.)

---

> ### Author Rebuttal · Authors · 2023-08-28
>
> Thank you for the helpful review!
>
> Regarding algorithmic complexity, the quadratic cost in sequence length n results from how we selected the number of routed tokens m as a function of n for simplicity of analysis.  For example, with fine-tuning we typically set m=1024 for sequence length n=16384 and therefore replace m with n/16.  In practice, when scaling to sequence length 64k for NarrativeQA for example, we actually found we could reduce m to n/32, so the algorithmic analysis that sets m as a constant fraction of n could be considered more of a “worst case” for further length scaling, and we’re optimistic that scaling can look better than this in practice, as often many practical long-context applications have an increasing fraction of less relevant tokens the longer the context window.
>
> Regarding training from scratch, this was unfortunately necessary due to the differences in the new light/heavy branches, but we’ve seen concurrent related work on conditional adapters (https://arxiv.org/abs/2304.04947) which we consider a promising future direction to “adapt” the light or heavy branch starting from an existing Transformer checkpoint.
>
> Thank you for the nice suggestion about Table 6 also.  We’ll be sure to highlight the best scores for easier comparison.

---

### Official Review · Reviewer_gfnU · 2023-08-06

**Soundness:** 4

**Excitement:**

4: Strong: This paper deepens the understanding of some phenomenon or lowers the barriers to an existing research direction.

**Paper Topic And Main Contributions:**

The authors present a transformer model designed for handling long-input texts. They split the computational process into two branches: a light one that processes all tokens and a heavy one that focuses on important tokens selected by a trainable routing network. By incorporating multi-query cross-attention and pretraining with UL2, they showcase the model's performance and efficiency in both few-shot and fine-tuning scenarios.

**Questions For The Authors:**

A. In the computation of FLOPs for the FFds, it's mentioned in lines 231 and 231 that r_L and r_H represent the dimension ratios relative to the standard T5. Considering this, it appears that the formula on line 235 should include squares for both r_H and r_L since they are applied to "d." If that adjustment is made, the entire FLOPs formulation and final FLOPs formulas would likely change, although this change should not affect the experimentation results.

B: Is it possible for this architecture to accommodate a more customized setting where each layer has its own ratio? For instance, the model could have larger local attention windows "w" and lower ratios "m" for the early layers, while the higher layers could have smaller local attention windows and higher important token ratios, considering their increased contextualization.

**Reasons To Accept:**

- The paper has a clear and well-organized structure, with a concise statement of objectives and clear description of the methodology.
- The idea of integrating both a light route and a heavy route using a dynamic routing mechanism is thoughtfully designed and appears to be an innovative approach, particularly for long-range sequence-to-sequence models.
- The extensive range of experiments conducted across various NLP tasks convincingly demonstrates the model's superior performance.
- The inclusion of a thorough ablation study and analysis on the routed tokens provides valuable insights into the design decisions and inner workings of the model.

**Reasons To Reject:**

My primary concern is that the selected important tokens ratio (m) in this mechanism remains static after pretraining. In some cases, particularly when training a multilingual version of this model, the ratio of important tokens could be dependent on the task/language. I wonder if it is possible to adjust this ratio during finetuning?

**Reproducibility:**

4: Could mostly reproduce the results, but there may be some variation because of sample variance or minor variations in their interpretation of the protocol or method.

**Reviewer Confidence:**

3: Pretty sure, but there's a chance I missed something. Although I have a good feel for this area in general, I did not carefully check the paper's details, e.g., the math, experimental design, or novelty.

**Typos Grammar Style And Presentation Improvements:**

A. It would be an improvement if some concepts have a brief description, like the UL2 objective and MQA.

B. If PEGASUS is weaker in a few-shot setup (according to lines 474-475) better demonstrate in some experimental results.

---

> ### Author Rebuttal · Authors · 2023-08-28
>
> Thank you for the helpful review and questions!
>
> To address your concern about the selected important token ratio, it is indeed possible to change this ratio during fine-tuning.  For example, for pre-training we used m=512 important tokens out of 4096 sequence length, yielding a ratio of 1/8.  On the other hand, for fine-tuning we used m=1024 important tokens by default out of 16384 sequence length for most tasks, yielding a smaller 1/16 ratio.  For our length scaling experiment for NarrativeQA, we actually found we only needed m=2048 tokens for 64k (more precisely 65,536) input length, yielding a ratio of 1/32.  We’ll clarify this in the text.  We agree it’s quite nice to be able to change this ratio depending on the task and language as you mentioned!
>
> Question A:
> Good question about dimensions.  The reason we do not square r_L or r_H is because we only apply this ratio to the feedforward “hidden dimensions”, not the original “model dimension”.  For example, for T5 or LongT5 Base, the model dimension is 768 while the feedforward hidden dimension is 2048.  For CoLT5 Base we keep the same 768 model dimension and use 1024 and 8096 for light and heavy feedforward hidden dimensions, respectively.  Table 7 in the appendix has more details for each model size, but perhaps we should move this up to the main paper.
>
> Question B:
> Great suggestion regarding different ratios for different layers.  The architecture can indeed accommodate this, and although we didn’t investigate such configurations in this paper, we think it’s an interesting future direction.  The routing analysis visualizations for TriviaQA did indicate that later layers may be doing more meaningful routing, so perhaps using relatively larger important token ratios at later layers would be beneficial as you suggest.
>
> Suggestion A:
> Thanks, we have some additional space we can use to elaborate a bit on UL2 and MQA in case of acceptance.
>
> Suggestion B:
> Performance was nearly 0 for few-shot using the PEGASUS pre-trained checkpoints, such that the metrics would not look interesting, but we can highlight this in the text.

---

### Official Review · Reviewer_tPfd · 2023-08-14

**Soundness:** 4

**Excitement:**

4: Strong: This paper deepens the understanding of some phenomenon or lowers the barriers to an existing research direction.

**Paper Topic And Main Contributions:**

* This paper deals with the problem of handling longer inputs with Transformers.
* Problem arises not only due to quadratic attention complexity but also because of  feedforward and attention layer projection to every token
* Authors build on top of LongT5 and suggest to divide each feedforward and attention layer into light branch and heavy branch and perform routing of tokens.
* Consists of Light branch and heavy branch
    * Light branch : caters to all tokens, has lower hidden dimension, performs local attention, has fewer heads than LongT5
    * Heavy branch : caters to important tokens, has higher hidden dimension, performs global attention, has more heads than LongT5
* Authors suggest learning routing function using learned embeddings.
* Apart from architectural change, authors also use a different objective for pre-training and add multi-query cross-attention to speed up inference
* CoLT5 is faster when it comes to inference or fine-tuning as compared to LongT5. Does better on almost all the datasets as compared to LongT5 and achieves SOTA results on SCROLLS benchmark
* As input length scales up, the model performance improves and it achieves faster speed than LongT5 (model scales well)


**Questions For The Authors:**

A. Apologies if I’ve missed that part but after going through the pre-training section (page 6) I realized the dataset on which CoLT5 is pre-trained isn’t mentioned.

B. Reproducibility - Numbers of CoLT5 -B in Table 6 and Table 3 are different. I wonder why that is happening. QAS F1 score is 42.1 v/s 38.3 in Table 3 and Table 6 respectively. Same for QMS and GovR Rgm scores have a difference of almost 5% which is a bit high. Let me know if I’m missing something that’s different in both these tables.

C. Are all these experiments done on multiple seeds because I couldn’t see that anywhere.

D. In Table 5, when you say the number of Natural Questions or TriviaQA, the number stated is in decimals like 0.1. Is it % or in hundreds, thousands? It will be great if you can clear that.

E. Why didn’t you pre-train CoLT5 on input length equivalent to that of the dataset with the highest median input length in Table 4? Was it because there were no significant gains after a certain input length or did it take too long?

F. I can’t see the number of parameters for different versions of CoLT5. The paper only mentions that it has more overall parameters as compared to LongT5 but uses less compute (Update: The concerned numbers are mentioned in Appendix, I’d recommend moving them in the main paper. Not the whole table but the parameter count)


**Reasons To Accept:**

* Motivation and intuition behind the paper are explained really well.
* Changes to the existing architecture of LongT5 are quite simple and novel.
* CoLT5 is not just doing well when it comes to numbers but also efficient as compared to the baseline mentioned.
* Conditional computing section is explained thoroughly except for Conditional Attention sub-section
* Utilizes SCROLLS benchmark precisely made for the task at hand covering summarization, question answering, and natural language inference tasks.
* Promising performance with extremely long inputs (~64k tokens). This is one of the main highlights of the paper for me.
* Ablation study is done quite well serving as a foundation for upcoming models dealing with large input length


**Reasons To Reject:**

* Results look great, however, SCROLLS is limited to English only. Testing on benchmarks focusing on other languages as well could have been helpful.
* Selection of light and heavy hidden dimensions (rL, rH) looks a bit hacky especially for calculating FLOPS for the CoLT5 attention layer  (rL=¼, rH=¾). An ideal scenario would have been to show trade off with different hidden dimensions, corresponding parameter count and show its effect on downstream tasks (apologies if it’s already mentioned anywhere)


**Reproducibility:**

4: Could mostly reproduce the results, but there may be some variation because of sample variance or minor variations in their interpretation of the protocol or method.

**Reviewer Confidence:**

4: Quite sure. I tried to check the important points carefully. It's unlikely, though conceivable, that I missed something that should affect my ratings.

**Typos Grammar Style And Presentation Improvements:**

* One minor correction: Page 6, “SCROLLS contains question-answering datasets: Narrative QA …, NLI dataset: ContractNLI.., and summarization datasets: SummScreenFD.. ”. Minor punctuation adjustments that can improve clarity and readability.
* “Feedforward and projection layer” and “feedforward and attention layer” have been used throughout the paper. I’d recommend sticking to one of those preferably feedforward and attention projection layers.
* Figure 2 doesn’t mention the metric of average perf which is F1 score. Also, the caption should be: CoLT5 achieves faster performance than LongT5 for the same number of parameters instead of saying CoLT5 achieves stronger performance than LongT5 at any speed.
* In Table 3 you have computed the average score under the Avg column, however, these metrics may not always be directly comparable, so averaging them might not accurately represent the overall performance. You should have averaged similar metrics like TQA, NQA, QAS under avg F1, QuAL, CNLI under avg EM, and arXiv, SumS, QMS, GovR under avg Rgm.
* I feel the equation for calculating FLOPs (attn), the global projection part, can be explained a bit. It took me some time to understand that (this one is totally up to you, if you think readers can figure this out easily then don’t)

---

> ### Author Rebuttal · Authors · 2023-08-28
>
> Thank you for the thorough review and engaging questions!
>
> We agree that CoLT5 evaluation on multilingual tasks would be very interesting too but unfortunately could not find an extensive suite of multilingual long input tasks like SCROLLS.
>
> Question A:
> Great question.  We pre-trained on the C4 dataset used by T5 (and LongT5).  We’ll be sure to mention this in the pre-training section.
>
> Question B:
> Thank you for the careful comparisons.  The explanation for the difference is that Table 3 uses official SCROLLS leaderboard test set results, while all ablations in Table 6 use validation sets instead.  Most of the SCROLLS datasets have a relatively modest number of training examples (10k or less), so we saw that the CoLT5 Base model “overfit” slightly with higher metrics on the validation set than the leaderboard test set for QMS and GovR as you noticed.  Interestingly we noticed less such “overfitting” for CoLT5 Large and even less for CoLT5 XL.  On the other hand, for the QAS dataset we found the validation metrics to be noticeably lower than leaderboard metrics in general, and we suspect this is due to the QAS test set having more examples with multiple references (98% in test vs. 74% in validation as reported in the original Qasper paper).
>
> Question C:
> We unfortunately lacked the resources to repeat fine-tuning for each dataset using multiple seeds and relied more on averaging _across_ the 9 datasets to compare overall performance with less noise.
>
> Question D:
> We’ll clarify the Table 5 caption to highlight that it’s showing the average number of few-shot examples that fit in context along with the example we’re answering the question for.  The 0.1 value for NQ at 1024 length for example indicates that very few examples (e.g. 1 in 10) have space to fit even 1 shot, so effectively 1024 length degenerates to zero-shot for most NQ examples.
>
> Question E:
> For comparison with LongT5, we wanted to make the pre-training recipe as similar as possible, so we stuck with LongT5’s 4096 input length for pre-training.  (Unlike LongT5, we did adopt the UL2 task, but this was necessary to yield any interesting few-shot metrics.)
>
> Question F:
> In case of acceptance we’ll have some extra space we can use to see if there’s a natural spot to add this in the main paper.
>
> Thank you also for all the helpful suggestions.  We’ll incorporate the grammatical clarifications and take everything into consideration in a final draft in case of acceptance.

---

### Meta-Review · Area_Chair_fnft · 2023-09-17

**Recommendation:** 5

**Metareview:**

The paper presents a conditional computational graph based on the importance of the tokens. Only a pre-set number of important tokens would be sent to heavy modules and thus the inference would be much faster.

Good parts:

1. The idea of using a light route and a heavy route with a dynamic routing mechanism is novel and interesting.

2. Experiments are solid to justify the claim.

3. The introduction motivates well.

Few potential issues:

1. Most experiments are performed on TPUs which limits its accessibility to mostly Googlers.

2. Description of method is mixed with many FLOP analysis, which is hard to digest. It's not very easy to understand what's the main proposed components and leave the FLOP analysis a separate session.

---

### Decision · Program_Chairs · 2023-10-07

**Decision:**

Accept-Main

**Comment:**

The paper presents a conditional computational graph based on the importance of the tokens. Only a pre-set number of important tokens would be sent to heavy modules and thus the inference would be much faster.

Good parts:

1. The idea of using a light route and a heavy route with a dynamic routing mechanism is novel and interesting.

2. Experiments are solid to justify the claim.

3. The introduction motivates well.

Few potential issues:

1. Most experiments are performed on TPUs which limits its accessibility to mostly Googlers.

2. Description of method is mixed with many FLOP analysis, which is hard to digest. It's not very easy to understand what's the main proposed components and leave the FLOP analysis a separate session.